# Towards Causal Replay for Knowledge Rehearsal in Continual Learning

**Nikhil Churamani**[1*], **Jiaee Cheong**[1*], **Sinan Kalkan**[2], **and Hatice Gunes**[1]

[1] Department of Computer Science and Technology, University of Cambridge, United Kingdom
[2] Department of Computer Engineering, Middle East Technical University, Turkey
{nikhil.churamani, jiaee.cheong, hatice.gunes}@cl.cam.ac.uk, skalkan@metu.edu.tr

### Abstract

Given the challenges associated with the real-world deployment of Machine Learning (ML) models, especially towards efficiently integrating novel information *on-the-go*, both Continual Learning (CL) and Causality have been proposed and investigated individually as potent solutions. Despite their complimentary nature, the *bridge* between them is still largely unexplored. In this work, we focus on causality to improve the learning and knowledge preservation capabilities of CL models. In particular, positing *Causal Replay* for knowledge rehearsal, we discuss how CL-based models can benefit from *causal interventions* towards improving their ability to replay past knowledge in order to mitigate forgetting.

## 1 Introduction

Real-world application of Machine Learning (ML) solutions require models to dynamically learn and adapt with streams of *incrementally* acquired data, while preserving past knowledge. Conventional ML-based methods are ill-fated to meet these challenges as they work under a pivotal assumption that all data is available a priori under relatively stationary data distributions (Graffieti, Borghi, and Maltoni 2022). This *stationarity* ensures that training samples are *independent and identically distributed* (i.i.d), allowing models to learn in batches of representative distributions. The real-world, however, is not stationary and changes continuously (Hadsell et al. 2020). As models continually encounter novel information, violating this *i.i.d* assumption, their ability to remember previously learnt tasks progressively deteriorates, resulting in forgetting (McCloskey and Cohen 1989).

Continual Learning (CL) (Parisi et al. 2019; Hadsell et al. 2020) aims to address adaptability in ML-based models by enabling them to *continually* learn and adapt, balancing *incremental learning* of novel information with the *preservation* of past knowledge. CL focuses on learning with continuous streams of data acquired from non-stationary or changing distributions (Hadsell et al. 2020). This may be achieved by *regulating* model updates to control plasticity or *rehearsing* past knowledge by storing and *replaying* already seen information to simulate *i.i.d* learning settings.

---

*These authors contributed equally.

Given the above, Causality (Pearl 2009), especially addressing adaptability and causal discovery, can complement lifelong learning of information by helping understand the causal structure of the data or the task and 're-adjust' model learning to cope with changing data distributions (Pearl 2019). Furthermore, it has been posited that the increasingly apparent challenges in ML (such as robustness, generalisation, bias, transparency) are due to conventional ML methods learning *correlation*-based patterns and relationships (Schölkopf et al. 2021). *Causal reasoning* tools can contribute towards understanding (Cheong, Kalkan, and Gunes 2023) and addressing some of these challenges (Cheng et al. 2022).

In this position paper, we focus on *knowledge rehearsal* as an effective tool for CL-based models to preserve past knowledge particularly using *causal interventions* to understand and update data distributions such that only the most relevant data samples (for *rehearsal*) or features (for *pseudo-rehearsal*) can be used by the model to preserve past knowledge. Such *Causal Replay* can help improve the efficiency of knowledge rehearsal for continual learning of information.

### 1.1 Knowledge Rehearsal to Mitigate Forgetting

Efficient rehearsal of past knowledge can be achieved by physically storing samples from previous tasks in memory buffers and regularly sampling from them (*rehearsal* – Robins 1993) mixing it with new data. The simplest strategy to achieve this is to fix the size of the memory buffer to be '*large enough*' and *randomly* maintain a fraction of previously seen samples from each task in the buffer for periodic rehearsal (Hsu et al. 2018). However, as the number of tasks increases, fewer samples are available for rehearsal per task. Other *sophisticated* rehearsal methods focus on prioritising replay following certain heuristics such as feature or classification margins (Hu, Zhang, and Zhu 2021), or storing exemplars for each task that best approximate task means (Rebuffi et al. 2017). Despite such 'intelligent' sampling, high dimensionality of data and a large number of tasks require a huge amount of memory, making their real-world application inefficient (Kwon et al. 2021).

Alternatively, generative models may be used, along with the learning model, that learn the inherent data statistics, enabling models to draw *pseudo-samples* to be replayed (*pseudo-rehearsal* – Robins 1995) along with novel data.

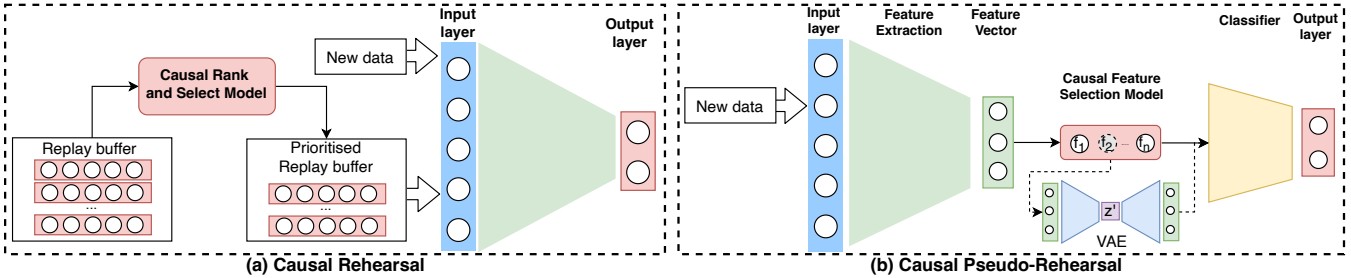

Figure 1: Causal Replay for (a) Prioritised Rehearsal and efficient (b) Pseudo-rehearsal of past knowledge.

Recent advances in generative models (Goodfellow et al. 2014; Kingma and Welling 2013), particularly in their ability to generate high-quality samples, have greatly enhanced the potential of pseudo-rehearsal methods (Shin et al. 2017; Churamani and Gunes 2020). More recent methods focus on *generative feature replay* (van de Ven, Siegelmann, and Tolias 2020; Stoychev, Churamani, and Gunes 2023), alleviating the need to optimise generators for reconstructing high-dimensional samples. However, as the number of tasks increases, they face *capacity saturation* and are not able to efficiently learn task-discriminative representations. Furthermore, the *generators* become harder to train, resulting in an inefficient rehearsal of past knowledge. We believe causality can offer significant improvements in this regard. To date, there is minimal work that explores the synergies between CL and causality (Chu, Rathbun, and Li 2021).

## 1.2 Causality

The study of causality entails a range of tools such as graphical models, the *do*-operator, counterfactuals as well as structural equations (Pearl 2009). Using these tools, conventional causal research has mostly focused on causal pattern recognition (Vowels, Camgoz, and Bowden 2021) and causal distribution estimation (Yao et al. 2021), Here, we focus on methods to merge conventional causal research with ML to address the existing gaps. Recent works in causal interpretability (Moraffah et al. 2020) and causal fairness (Makhlouf, Zhioua, and Palamidessi 2020) have proven such an approach to be promising. Here, we leverage two main themes: *Causal Interventions* and *Causal Structure Discovery*.

Following Pearl's notation (Pearl 2009) for a Structural Causal Model (SCM), we have a set of variables $V$ and a set of functions $F$ that encode the causal relations between each variable. Using this framework, *causal interventions* can be either be 'structural' or 'parametric' (Spirtes et al. 2000) representing a continuum of 'harder' to 'softer' interventions. A 'hard' intervention can be understood as a forcible removal of an edge such that the function encoding $V_i \leftarrow f_{V_i}$ is modified such that another variable $V_j$ is no longer a parent of variable $V_i$. 'Soft' interventions, on the other hand, simply modify the conditional probability distributions of the *intervened* variable $V_i$. Depending on the task, we can combine the most appropriate form of causal interventions with CL-based models to preserve past knowledge and update the model using only the relevant features. In ad-

dition, we also propose to leverage existing causal discovery methods (Vowels, Camgoz, and Bowden 2021) that can be utilised to discover causal relations within the observational data. We propose to impart the discovered causal knowledge to CL-based methods in order to mitigate forgetting and to learn new relevant features.

## 2 Causal Replay for Knowledge Rehearsal

Understanding the causal structure of the data can enable models to distil task-relevant information, positively impacting performance (Deng and Zhang 2021; Yang et al. 2021). This can either be in the form of identifying and prioritising data samples that contribute the most towards the models' learning (motivating *causal rehearsal*) or pruning feature-sets to extract meaningful representations that best attribute the task to be learnt (motivating *causal pseudo-rehearsal*). These possibilities are discussed further in this section.

## 2.1 Causal Rehearsal

One strategy for augmenting CL with causality can be causality-driven rehearsal (see Figure 1 a). Firstly, we aim to understand the causal structure of task-specific data in order to *prioritise* samples for rehearsal. As neural networks are capable of representing the input features as well as their respective causal relations to each other within their parameters, we can learn the causal structure of the data during the training phase using a range of existing causal discovery methods (Vowels, Camgoz, and Bowden 2021). An online example of doing so is exemplified by Javed, White, and Bengio (2020) who propose to measure the variance in the weights of the model, over time, with spurious features resulting in high variance. As we are only able to discover causal Directed Acyclic Graphs (DAGs) up to Markov equivalence, we can subsequently leverage causal-scoring methods (Glymour, Zhang, and Spirtes 2019) or causality-based feature selection methods (Yu et al. 2020) to prioritise those samples for rehearsal that are deemed to have a higher causal effect on the target prediction outcome. In addition, given the causal structure of the data, we can therefore prune off features that have the least or weakest causal effect on the target outcome. Subsequently, as we update the model with each new task, we can *reprioritise* the samples to update the memory buffer as well as the learnt causal structure. As such, the causal model can then also be updated in a continual manner as more data becomes available.

## 2.2 Causal Pseudo-rehearsal

Another opportunity is that of causality-driven pseudo-rehearsal (see Figure 1 b). Here the goal is to use the learnt causal structure of the data to rehearse information in a principled manner. Attempts to remove unwanted causal relations has proven to be effective in the case of knowledge distillation (Deng and Zhang 2021). However, such an idea has yet to be fully explored in CL. Existing methods largely rely on pattern generation to simulate i.i.d. settings. However, this does not take into account the causal relations between variables. One way of addressing this is to make use of interventions (both 'hard' and 'soft') such that we generate samples from the updated distribution which has been 'intervened' upon. Such an approach has proven to be effective in the domain of disentangled representation learning using Variational Autoencoders (VAEs) (Yang et al. 2021). Instead of simply generating pseudo-samples, we can intervene by updating the parameters of the generative model based on the causal effect estimated or parameterised by the learnt causal structure of the data. These parameters can also be continually updated given new information. By conducting pseudo-rehearsal in this manner, we are able to adapt to the changes in new data whilst preserving old information.

## 3 Summary and Next Steps

In this position paper, we propose to learn the causal structure of the data for efficient knowledge rehearsal in CL models. Understanding causal relationships can help distil knowledge by prioritising samples that contribute most towards model learning (causal rehearsal) as well as prune feature-sets to include only the most relevant features (causal pseudo-rehearsal), having the strongest causal relationships vis-à-vis the tasks at hand. Yet, as causal relations can be problem or task-specific (as opposed to task-agnostic), it will also be important to consider the causal relationship and dependencies between the different tasks to be learnt, across datasets. Some possible directions may involve exploring image datasets such as ImageNet (Deng et al. 2009) and CIFAR-10 (Krizhevsky and Hinton 2009) where models need to learn different objects or, more Facial Expression Recognition (FER) datasets such as Affect-Net (Mollahosseini, Hasani, and Mahoor 2018) and RAF-DB (Li, Deng, and Du 2017) where the models need to learn to classify different facial expressions. FER-based applications can be particularly interesting to explore given the high overlap in the learnt feature-spaces for the different tasks (evaluating human faces for expression recognition) as well as subject-specific variations in data samples. Thus, FER benchmarks will form the pivotal focus of our further experimentation with *Causal Replay*.

## Acknowledgement

For the purpose of *open access*, the authors have applied a Creative Commons Attribution (CC BY) license to any Accepted Manuscript version arising.
**Funding Details:** N. Churamani and H. Gunes are supported by the EPSRC/UKRI Project ARoEQ under grant ref. EP/R030782/1. J. Cheong is supported by the Alan Turing Institute Doctoral Studentship and the Cambridge Commonwealth Trust.

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
