# OpenReview forum: "Towards Causal Replay for Knowledge Rehearsal in Continual Learning"
_AAAI.org/2023/Bridge/CCBridge — AAAI23 Bridge Continual Causality_

### Official Review · Reviewer_o3mo · 2022-11-24
**Exploiting causal relationships in the replay buffer**

**Rating:** 7
**Confidence:** 4

**Review:**

The authors propose to augment replay strategies in continual learning with causal models. One proposal is to use the causal model to provide a "score" measuring the importance of the replay sample, thus allowing to order the replay buffer based on the score. Another proposal is related to generative replay. The authors suggest to remove unwanted causal relationships in the generated samples through intervention to improve the replay performance.
It is not entirely clear how the causal model can provide a relevant score for the former case. The generative replay case seems more interesting, even though it is not clear how to identify the "unwanted" causal relationships to be removed from the generated samples.

---

### Official Review · Reviewer_vCPS · 2022-12-01
**Promising proposal that uses tools from causality research to improve continual learning through replay**

**Rating:** 8
**Confidence:** 4

**Review:**

- This paper proposes to use insights from research on causality to improve the way in which knowledge rehearsal operates in a continuous learning setting.
- The authors propose two methods of causal replay: prioritized rehearsal (where interventions are used to understand the causal structure of the domain, and then prioritize replays in light of this causal knowledge), and causal-pseudo rehearsal (where knowledge of the causal structure is used for encoding new data in a way that distills the causally relevant information).
- These methods strike me as innovative and the authors do a good job laying them out in this short format.
- In cognitive science, there is a lot of work showing that memory is not only replay but also construction (i.e. our current knowledge affects how we recall what happened). It would be interesting to explore methods that don't just re-prioritize the replay of existing episodes, but where what is being replayed is affected by current causal knowledge (which could lead to episodes being replayed that didn't actually happen that way but are consistent with the current causal knowledge).

---

### Official Review · Reviewer_Qtmj · 2022-12-05
**Good ideas, even though details weren't clear to me.**

**Rating:** 9
**Confidence:** 5

**Review:**

This paper attempts to improve replay for CL using ideas from causality. Two ideas are presented. First one is to prioritize relevant past data using "Causal rank and select model", and the second idea is by "updating the parameters of the generative model based on the causal effect estimated or parameterised by the learnt causal structure of the data."

Overall, I like what the paper is proposing, that is, to use ideas from causality to improve CL. Theory suggests that such ideas should be useful, but I haven't seen much on this topic. I couldn't understand many things, but perhaps they are because the paper is short. But I will write about them anyways,
- Clarify the methods discussed (causal rehearsal and causal pseudo-rehearsal)
- Clarify why these methods will be better than other non-causal methods
- What kind of benchmarks one can test these theories on?

To help the authors I will copy past some parts of the paper that I really like. For the future writing, this may be helpful to focus the story
- Given the above, Causality (Pearl 2009), especially ad- dressing adaptability and causal discovery, can comple- ment lifelong learning of information by helping under- stand the causal structure of the data or task....
- Furthermore, it has been posited that the increasingly apparent challenges in ML (such as ro- bustness, generalisation, bias, transparency) are due to con- ventional ML methods learning correlation-based patterns and relationships (Scho ̈lkopf et al. 2021). Causal reasoning tools can contribute towards addressing some of these chal- lenges (Cheng et al. 2022).
- ... such that only the most relevant data samples (for rehearsal) or features (for pseudo- rehearsal) can be used by the model to preserve past knowl- edge.
- We propose to impart the discovered causal knowledge to CL-based methods in order to mitigate forgetting and to learn new relevant features.
- ... we can learn the causal structure of the data during the training phase using a range of existing causal discovery methods (Vowels, Camgoz, and Bowden 2021).
- we can intervene by updating the parameters of the generative model based on the causal effect estimated or parameterised by the learnt causal structure of the data. These parameters can also be continually updated given new information.

Here is a sentence I didn't like in the abstract. I also think that the paper is very well written but the abstract, not so much.
- ...  both Con- tinual Learning (CL) and Causality have been proposed and investigated individually as potent solutions

Good luck with your future submissions!

---

### Decision · Program_Chairs · 2022-12-05

**Decision:**

Accept

**Comment:**

Accept - Oral

This paper proposes using ideas from causality research to improve replay methods for continual learning. The reviewers agree that the ideas are interesting, and the topic is a great fit for the bridge program. We suggest that the authors use the additional space in the camera-ready version to integrate the reviewers’ comments and concerns, including clarifications and a discussion on potential benchmarks.